# 4-Mercaptopyridine-Modified Sensor for the Sensitive Electrochemical Detection of Mercury Ions

**DOI:** 10.3390/mi14040739

**Published:** 2023-03-27

**Authors:** Mingjie Han, Yong Xie, Ri Wang, Yang Li, Chao Bian, Shanhong Xia

**Affiliations:** 1State Key Laboratory of Transducer Technology, Aerospace Information Research Institute, Chinese Academy of Sciences, Beijing 100190, China; 2School of Electronic, Electrical and Communication Engineering, University of Chinese Academy of Sciences, Beijing 100190, China

**Keywords:** electrochemical sensor, molecular simulation, binding energy, 4-mercaptopyridine, mercury detection

## Abstract

As a highly toxic heavy metal ion, mercury ion (Hg^2+^) pollution has caused serious harm to the environment and human health. In this paper, 4-mercaptopyridine (4-MPY) was selected as the sensing material and decorated on the surface of a gold electrode. Trace Hg^2+^ could be detected by both differential pulse voltammetry (DPV) and electrochemical impedance spectroscopy (EIS) methods. The proposed sensor displayed a wide detection range from 0.01 μg/L to 500 μg/L with a low limit of detection (LOD) of 0.002 μg/L by EIS measurements. Combined with molecular simulations and electrochemical analyses, the chelating mechanism between Hg^2+^ and 4-MPY was explored. Through the analysis of binding energy (BE) values and stability constants, 4-MPY showed an excellent selectivity for Hg^2+^. In the presence of Hg^2+^, the coordination of Hg^2+^ with the pyridine nitrogen of 4-MPY was generated at the sensing region, which caused a change in the electrochemical activity of the electrode surface. Due to the strong specific binding capability, the proposed sensor featured excellent selectivity and an anti-interference capability. Furthermore, the practicality of the sensor for Hg^2+^ detection was validated with the samples of tap water and pond water, which demonstrated its potential application for on-site environmental detection.

## 1. Introduction

Heavy metal ions are one of the important contributors to environmental pollution [1]. When the concentration of heavy metal ions exceeds the limit of tolerance, it can cause damage to the ecological environment and harm to human health. As a source of heavy metal pollution, mercury contamination exists mainly as divalent mercury ions (Hg^2+^) in water environments. Due to the bioaccumulation effect, Hg^2+^ can be enriched in the human body through the food chain, causing permanent damage to the nervous, cardiovascular and skin systems even at very low levels [2,3,4]. So far, the methods for Hg^2+^ detection mainly include colorimetry [5], fluorescent probes [6], surface-enhanced Raman chips [7], fiber-optic sensing techniques [8], etc. There is still a great need to develop new simple methods for Hg^2+^ detection with high sensitivity, strong selectivity and low costs.

Electrochemical methods, which are commonly used for chemical [9,10] and biological [11,12] detection, have attracted increasing attention, due to their good sensitivity, low detection limits and simple operation. According to the WHO requirement, the limit for Hg^2+^ in drinking water is 6 μg/L, which poses a great challenge to the sensors [13]. To improve the performance of the electrochemical sensors, plenty of novel materials have been proposed and applied as the sensing elements, including metal–organic frameworks (MOFs) [14,15,16], graphene [17,18,19], carbon nanotubes [20,21,22], oligonucleotides [23,24,25], etc. For instance, Mariyappan et al. reported the electrochemical detection of Hg^2+^ using a nanocomposite of a Sr@FeNi-S nanoparticle and carbon nanotube with the limit of detection (LOD) of 0.1 μg/L and a wide linear range of 10 μg/L–55.8 mg/L [26]. Teodoro and co-workers modified a hybrid nanocomposite composed of electrospun nanofibers containing cellulose nanowhiskers and reduced graphene oxide onto the surface of an electrode to analyze Hg^2+^ and the LOD was 1.04 μg/L [27]. Fu et al. fabricated a novel-type sensor of Zr (IV)-based MOFs to realize the detection of Hg^2+^ via a T-Hg^2+^-T hairpin with an LOD of 1.46 pg/L [28]. While these methods have achieved good results, they typically require complex preparation processes or expensive chemical materials. In order to achieve low-cost methods for the analysis of heavy metals, Scandurra et al. proposed a disposable and low-cost electrode based on a graphene paper-nafion-Bi nanostructures material for the simultaneous determination of cadmium and lead [29]. Moreover, Hu et al. reported a method for copper determination using a composite prepared from N-doped carbon spheres/multi-walled carbon nanotubes with an LOD of 0.092 μg/L [30].

In this work, we have introduced 4-mercaptopyridine (4-MPY) as a sensing material for Hg^2+^ detection. Among the various types of N, pyridinic-N is more active compared to the other types of N, because it can induce sufficient defects and more active sites, and is more energetically favorable towards adsorbing alkali metal ions [31,32]. As a bifunctional molecule, 4-MPY can be self-assembled at the surface of a Au electrode through the formation of a Au–S bond and its coordination with Hg^2+^ via the nitrogen of the pyridine moiety to form a Hg(pyridine)_2_ (Hg(py)_2_) complex [33,34]. Figure 1 illustrates the principle of the proposed sensor. The geometric structure of the complex is optimized by the density functional theory (DFT) calculation and the binding energies (BE) of different ions are calculated. In the presence of Hg^2+^, Hg(py)_2_ will accumulate at the sensing region, which leads to a reduction in the electrochemical activity of the electrode surface. The concentration of Hg^2+^ can be detected by analyzing the variation of current or impedance. This proposed electrochemical sensor exhibits excellent sensitivity, selectivity, a wide detection range and low LOD. In addition, this sensor has been employed to detect trace Hg^2+^ in water environments via the standard addition method with satisfactory recovery.

## 2. Materials and Methods

### 2.1. Materials and Apparatus

The molecule 4-MPY was purchased from Aladdin Chemistry Co., (Shanghai, China). Ethylenediamine tetraacetic acid (EDTA), chloroauric acid (HAuCl_4_·4H_2_O), potassium ferricyanide (K_3_[Fe(CN)_6_]), potassium ferrocyanide trihydrate (K_4_[Fe(CN)_6_·3H_2_O]), MgCl_2_, CaCl_2_, KCl and acetone were purchased from Sinopharm Chemical Reagent Co., (Shanghai, China). Different interferents including Hg^2+^, Cu^2+^, Pb^2+^, Zn^2+^, Se(IV) and Sn^2+^ were bought from the Institute for Environment Reference Materials Ministry of Environmental Protection (Beijing, China). The actual water samples were obtained from our lab tap and Tsinghua Lotus Pond, respectively. All the solutions were prepared by deionized water (18.2 MΩ∙cm) produced by Milli-Q system (Millipore Company, Germany). All chemical reagents were used without any additional purification.

All the electrochemical measurements were performed with a Gamry Reference 600 electrochemical workstation (Gamry Instruments, Warminster, PA, USA). Cyclic voltammetry (CV), electrochemical impedance spectroscopy (EIS) and differential pulse voltammetry (DPV) measurements were performed by a three-electrode system consisting of a gold electrode as the working electrode, a platinum electrode as the counter electrode and a Ag/AgCl (3 M KCl, aq) electrode as the reference electrode. X-ray photoelectron spectroscopy (XPS) spectra were used to analyze the elements of the sensing surface in the vacuum condition by an ESCALAB 250Xi (ThermoFisher-VG Scientific, Waltham, MA, USA). The Hg^2+^ concentrations of the tap water and pond water samples were determined through the inductively coupled plasma mass spectrometry (ICP-MS) method by the PONY Testing International Group (PONY Company, Beijing, China).

### 2.2. Preparation of 4-MPY-Modified Gold Electrode

In this work, the sensing region of the electrochemical sensor was modified via the self-assembly method. The bare gold electrode was firstly ground and cleaned, then it was scanned with H_2_SO_4_ (0.5 M) and potassium ferricyanide (5 mM) successively to ensure that the current was roughly the same each time. Then the cleaned electrode was immersed in 4-MPY (1 mM) ethanolic solution overnight so that a packed 4-MPY monolayer film could be formed on the surface of electrode. Finally, the electrode was rinsed three times with ethanol and deionized water, successively.

### 2.3. Computational Method and Models of Hg(pyridine)_2_ Complex

All DFT calculations were completed using DMol^3^ with the generalized gradient approximation (GGA) scheme and the Becke-Lee-Yang-Parr (BLPY) exchange-correlation function to optimize the geometric structures of the compounds and vibration analyses. Atomic basis sets were applied numerically in terms of the double numerical plus polarization function (DNP) basis set, version 3.5. The core electrons were modeled using a DFT semi-core pseudopots (DSPP) method for metal cations. The DMol^3^ geometry optimization convergence tolerances of the energy, force, and displacement were 1 × 10^−5^ Hartree (Ha), 2 × 10^−3^ Ha/Å, and 5 × 10^−3^ Å, respectively.

### 2.4. Electrochemical Measurements

The binding process of the Hg^2+^ to the modified electrode surface was characterized by CV and EIS measurements, respectively. The CV measurements were performed over a potential range from −0.2 V to 0.6 V at a scan rate of 50 mV/s. The EIS measurements were carried out in the frequency range from 10^6^ Hz to 0.1 Hz in a galvanostatic mode with an alternating current (AC) amplitude of 50 mV. The DPV scanning was performed from −0.2 V to 0.6 V using a step size of 5 mV, a pulse size of 20 mV, a pulse time of 0.02 s and a sample period of 0.05 s. The substrate solutions contained 0.1 M KCl with 5 mM [Fe(CN)_6_]^3−/4−^ as a redox probe. 

Various concentrations of Hg^2+^ were prepared by a stock solution of Hg^2+^ solution (1 g/L). The DPV and EIS measurements were respectively performed for the Hg^2+^ detection after the modified electrode had been immersed in the test solution for 10 min. The reutilization of the sensor was investigated by immersing the sensor into the EDTA solution (1 mM) for 1 h. To study the selectivity and the ability to counter interference, different interferents, including K^+^, Mg^2+^, Ca^2+^, Cu^2+^, Pb^2+^, Zn^2+^, Se(IV) and Sn^2+^, were added separately to the test solutions. In addition, the practicability of the sensor was validated by spiking different concentrations of Hg^2+^ into the tap water and pond water samples.

## 3. Results and Discussion

### 3.1. Electrochemical Performance Characterizations of 4-MPY-Modified Electrode

In order to evaluate the charge transfer properties of the 4-MPY-modified electrode, the CV and EIS measurements were used to characterize each step of the modification, respectively. Figure 2a shows the CV results for the 4-MPY/Au after the reaction with Hg^2+^ (50 μg/L), 4-MPY/Au, and bare Au electrode in the presence of a 5 mM [Fe(CN)_6_]^3−/4−^ redox probe. The 4-MPY-modified sensor exhibited a lower voltammetric peak, indicating that the electron transfer ability on the electrode surface becomes poor. After the reaction with Hg^2+^, the current became significantly smaller. Figure 2b shows the corresponding EIS results for each step. In the impedance diagram, the diameter of the semicircle represents the charge-transfer resistance (R_ct_) of the electrode, which is a key parameter for studying the electron transfer barrier and surface interface interaction [35]. The EIS parameters were obtained by fitting the experimental data appropriately to a Randles equivalent circuit as shown in Appendix A, where a smaller diameter implies a smaller R_ct_ and a higher conductivity. The R_ct_ values were estimated to be 125.8 Ω, 685.7 Ω and 812.6 Ω for bare Au, 4-MPY/Au and 4-MPY/Au after their reactions with Hg^2+^, respectively.

In order to investigate the full redox process and better understand the changes in the electrochemical environment occurring at the electrode surface after the reaction with Hg^2+^ (10 μg/L), the CV measurements were performed. Figure 3a shows the results for different scanning rates. The intensity of the current gradually increased with the increase in the scan rate. As shown in Figure 3b, both the oxidation peak and reduction peak currents showed linear relationships along with the square root of the scan rate (v^1/2^), demonstrating a diffusion-controlled process.

### 3.2. XPS Characterizations of 4-MPY/Au

To further investigate the reaction mechanism between 4-MPY and Hg^2+^, the corresponding XPS spectra were thoroughly investigated. Figure 4a shows the survey spectra of the 4-MPY/Au electrode, indicating the presence of C, S, N and Au. After the reaction with Hg^2+^, another peak of Hg 4f_7/2_ was apparent at 99.97 eV [36], as shown in Figure 4b. Using the peak fitting method of Figure 4c, we found that a characteristic peak of S 2p_3/2_ was at 163.1 eV, which demonstrated the formation of Au–S bonding [37]. Furthermore, Figure 4d shows the deconvolved spectrum of N 1s. In contrast to the 4-MPY/Au electrode, a different single fitting peak appeared at 397.85 eV after the reaction with Hg^2+^, suggesting that the formation of this type of nitrogen element may have been related to the complexation of 4-MPY with Hg^2+^ [36,38]. The 4-MPY bound the Au surface via a mercapto group to form a monolayer film and coordinated Hg^2+^ via the nitrogen of pyridine to form the Hg(py)_2_ complex. Due to the presence of a pair of unbound isolated electrons of pyridine nitrogen in the sp^2^ hybrid orbital, the N atom exhibited a certain nucleophilic property and was capable of interacting with metal cations to form a metal–pyridine complex. The structural analyses of the 4-MPY/Au electrode confirmed the successful modification of 4-MPY at the Au electrode surface and provided a preliminary validation of the feasibility of the sensing principle for Hg^2+^ detection.

### 3.3. Computational Method and Models of Pyridine Complex

From the results of the electrochemical characterization, we found that the modification of the 4-MPY led to a deterioration in the electrochemical activity at the electrode surface. When the modified electrode was in contact with Hg^2+^, the current signal became smaller. To confirm this result, DFT calculations were performed using DMol^3^ with the GGA-BLPY functional to optimize the geometry of the metal cation-4-MPY (M-Py) complex.

The electrostatic potential diagram can reflect the electrical distribution of molecules. As shown in Appendix A, for 4-MPY, the negatively charged region was mainly concentrated near the pyridine nitrogen atom. Due to the nucleophilicity of the pyridine nitrogen, the exposed N atom of 4-MPY modified on the electrode surface would interact with the metal ions in the solution.

In addition, a model of the decomposition reaction of the M-Py compound was constructed to study the interaction between metal ions and 4-MPY, as shown in Figure 5. The BE values were calculated and the variation in the bond lengths between the metal cations and pyridine nitrogen were compared. Based on the results of the geometry optimization and vibration analyses, the BE could be obtained as follows:
(1)$$BE=E_{\mathrm{M-Py}}-E_{M}-E_{Py},$$
where $E_{\mathrm{M-Py}}$, $E_{M}$ and $E_{Py}$ are the energies of the M-Py compound, metal cation and 4-MPY, respectively. The BE values of different interferents were calculated and shown in Appendix A. For instance, for K^+^ and Hg^2+^, through the model analyses and DFT calculations, we found that the bond lengths of K^+^-Py and Hg^2+^-Py were 2.725 Å and 2.001 Å, respectively, while the BE values were determined to be −24.1 kcal/M and −322.2 kcal/M, respectively. It is well known that both shorter bonds and larger BE values can lead to stronger binding, which suggests that Py is more likely to be coordinated with Hg^2+^ rather than K^+^ when Hg^2+^ is present in the solution. 

According to Appendix A, compared with other metal ions, we found that the BE value between Hg^2+^ and 4-MPY was much larger, except for Cu^2+^ and Zn^2+^, indicating that Hg^2+^ probably has stronger binding ability with 4-MPY than other metal ions. Furthermore, the stability constants of M^2+^-pyridine (M = Cu^2+^, Zn^2+^ and Hg^2+^) complexes have been investigated and listed in Appendix A. According to the definition of stability constants, for the same type complexes with the same number of ligands, the larger the logK value, the greater the tendency to form a coordination ion and the more stable the complex is. As shown in Appendix A, it can be found that the stability constants of Hg^2+^-pyridine complexes was much larger than those of Cu^2+^-pyridine and Zn^2+^-pyridine [39,40,41], proving that 4-MPY was more likely to complex with Hg^2+^. In conclusion, we considered that 4-MPY had a strong selectivity for Hg^2+^ through the analyses of the BE values and stability constants.

As shown in Figure 6, in the presence of Hg^2+^, the intersection angle of C-S-Au became slightly larger, from 105.639° to 111.709°, which indicated that 4-MPY would be more perpendicular to the adsorption on the Au surface after the pyridine nitrogen atom complex with Hg^2+^. Therefore, when the 4-MPY molecules were self-assembled on the surface of the Au electrode, a high coverage of 4-MPY film could be obtained on the electrode surface by π–π stacking. This packed film prevented the diffusion of Hg^2+^ onto the surface of the gold electrode, which facilitated the chelation of Hg^2+^ with the N atoms of adjacent pyridine molecules and resulted in the 4-MPY-Hg^2+^-4-MPY complex as the primary form [42,43].

### 3.4. Optimization of Experimental Conditions

To improve the performance of the sensor, the concentration of 4-MPY was optimized. Because the current signal gradually decreased as the Hg^2+^ concentration increased, a large response current intensity was required when detecting in the substrate solution. Figure 7a shows the results of the optimization for the concentration of 4-MPY. With the increase in the 4-MPY concentration, the current intensity increased at first, then gradually tended towards being stable and reached its maximum value at 1 mM. Thus, 1 mM was chosen as the optimized concentration of 4-MPY.

To realize the rapid detection of Hg^2+^, the reaction time with the Hg^2+^ was also investigated. Figure 7b shows that the current variation continues to increase as the reaction time increases from 1 min to 30 min, but the increase became much smaller after 10 min. Thus, we chose 10 min as the reaction time for subsequent experiments, considering the time efficiency of the electrochemical detection.

### 3.5. Analytical Performance for Hg^2+^ Detection

The modified electrode was immersed in the solutions containing different concentrations of Hg^2+^. Using the optimized conditions, the performance of the modified sensor was investigated using DPV and EIS measurements, respectively. Figure 8a shows the results of the DPV current response for different Hg^2+^ concentrations, where we could obviously find that the current decreased gradually as the Hg^2+^ concentration increased. As shown in Figure 8b, the current variation increased as a function of the logarithm with Hg^2+^ concentration over the range of 0.1 μg/L to 500 μg/L with an excellent correlation coefficient of 0.994. Figure 8c shows that there was a good linear relationship between the current variation and lg C as well. The limit of detection (LOD) of the 4-MPY-modified gold sensor was calculated using the formula:
(2)LOD=3σ/s,
where σ is the standard deviation of seven times the DPV measurements for the blank solution and s is the slope of the calibration curve. The proposed sensor exhibited a low LOD of 0.02 μg/L, which was quite lower than the WHO requirement (6 μg/L) for Hg^2+^ in drinking water and indicated that the sensor was sensitive enough to assess the level of pollution.

During the process of the reaction, the formation of the complex on the electrode surface made the transfer of [Fe(CN)_6_]^3−/4−^ on the electrode surface difficult. We further investigated the performance of the modified sensor using the EIS measurement. The corresponding impedance diagrams are shown in Figure 9. A good logarithmic relationship was also exhibited. In addition, we found that the sensor could get a wider detection range between 0.01 μg/L and 500 μg/L and a lower LOD of 0.002 μg/L with the EIS measurement compared to the DPV measurement. At this point, the impedance detection method would be more sensitive than the conventional current detection method. Some of the recently reported electrochemical Hg^2+^ detection systems are listed in Table 1. By comparison, the 4-MPY-based sensor exhibited a wide range and a low LOD.

### 3.6. Repeatability, Reproducibility and Reusability Studies

The repeatability of the sensor was evaluated by performing seven consecutive experiments with the same electrode after the reaction with Hg^2+^. Appendix A shows the results of the 4-MPY-modified sensor for seven measurements under the same conditions, and the results show that there was a tiny relative standard deviation (RSD) of 0.63% between these independent experiments, which indicates a significant repeatability of the proposed sensor.

The reproducibility of the sensor was determined by preparing electrodes and using them for the determination of Hg^2+^. As shown in Appendix A, the current responses floated from 6.68 μA to 7.73 μA with a relatively small RSD of 4.8%, which demonstrates that the sensor has appreciable reproducibility.

To investigate the reusability of the prepared sensor, EDTA was used to remove the Hg^2+^ from the Hg(py)_2_ complex and the corresponding results are observed in Appendix A. As a chelating agent, EDTA has the ability to chelate metallic ions at a ratio of 1:1 [49,50]. After the reaction with Hg^2+^ or treatment with EDTA, the DPV measurements were performed on the sensor. It was found that the DPV current could be recovered to approximately 96.19% of the initial level, and then the sensor was used to detect trace Hg^2+^ again. It was found that the sensor can be reutilized, which shows a potential for the implementation of on-site detection.

### 3.7. Selectivity and Anti-Interference Ability Studies

Due to the specific coordination between Hg^2+^ and 4-MPY, the sensor showed a good selectivity and anti-interference ability [7,51]. Several common interferents, such as Ca^2+^, Sn^2+^, Se(IV), Mg^2+^, Pb^2+^, Zn^2+^ and Cu^2+^ were added separately into test solutions to investigate the selectivity of the 4-MPY-modified sensor. The concentrations of these interfering substances were 100 μg/L, which was 10 times larger than the concentration of Hg^2+^ (10 μg/L). As shown in Figure 10a, the 4-MPY-modified sensor had a strong response to Hg^2+^ compared with other heavy metal ions. In addition, these eight metal ions (100 μg/L) were respectively mixed with Hg^2+^ (10 μg/L) to assess the effects of interfering substances on the detection of Hg^2+^. As shown in Figure 10b, the response currents were altered slightly from 93.1% to 107.4% compared with the test solution containing Hg^2+^ only. As a result, the prepared sensor exhibited excellent selectivity and showed a good anti-interference ability when detecting Hg^2+^ in the water samples.

### 3.8. Detection of Hg^2+^ in Real Samples

To investigate the practical application of the 4-MPY-modified sensor, Hg^2+^ was detected in the tap water and pond water samples using the standard addition method, and the recoveries were calculated. After pretreatment of the original water samples, no Hg^2+^ was found by the ICP-MS method. Then three spiked samples with different concentrations of 0.01 μg/L, 0.05 μg/L and 10 μg/L were prepared by adding a Hg^2+^ standard solution into the water samples. Table 2 shows the results of the Hg^2+^ assay in the water samples with satisfactory recoveries ranging from 92.02% to 118.68%, which confirmed that the 4-MPY-modified sensor is suitable for the determination of Hg^2+^ in real samples.

## 4. Conclusions

In this work, a high-performance and easily fabricated electrochemical sensor for the detection of trace Hg^2+^ was developed. As a bifunctional molecule, 4-MPY can be self-assembled at the surface of Au electrodes through the forming of a Au–S bond and coordinated with Hg^2+^ via nitrogen of the pyridine moiety to form a Hg(py)_2_ complex, which leads to the deterioration in the electrochemical activity on the electrode surface. The chelating mechanism between Hg^2+^ and 4-MPY has been explored by molecule simulations and electrochemical characterizations. Through the analyses of the BE values and stability constants, we found that 4-MPY has an excellent selectivity for Hg^2+^. Under optimized conditions, trace Hg^2+^ can be detected by both the DPV and EIS measurements. The sensor exhibited a wider detection range from 0.01 μg/L to 500 μg/L with an LOD of 0.002 μg/L via the EIS measurement. Moreover, due to the strong specific binding capability between Hg^2+^ and 4-MPY, the sensor showed a high selectivity and strong anti-interference ability for Hg^2+^ with good repeatability and reproducibility. After treatment with EDTA, the electrode exhibited a potential to be reused. In addition, the sensor was also employed to detect Hg^2+^ in the real water samples with satisfactory recoveries, demonstrating the application potential of the sensor for on-site environmental detection.

## Figures and Tables

**Figure 1 micromachines-14-00739-f001:**
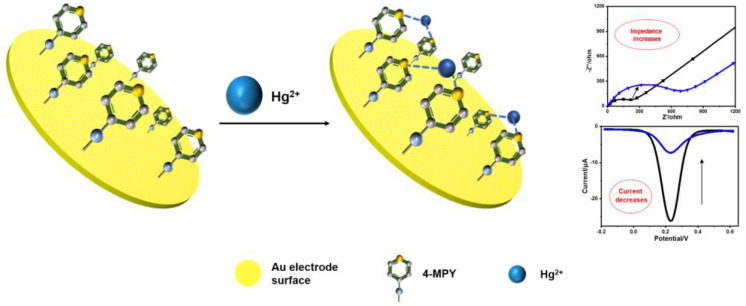
Schematic illustration for the Hg^2+^ detection process of 4-MPY-modified Au electrode.

**Figure 2 micromachines-14-00739-f002:**
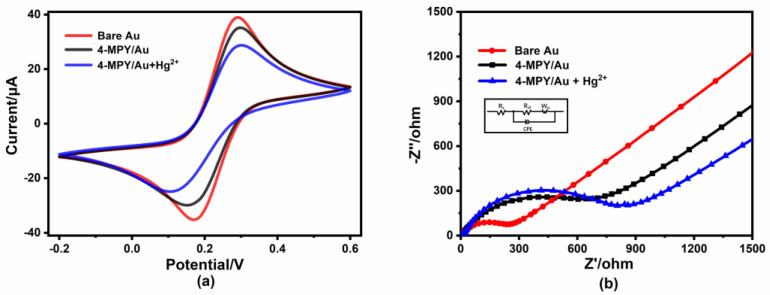
Electrochemical performance characteristics of Au electrode surface under different modification conditions. (**a**) CV curves, (**b**) EIS curves of bare Au, 4-MPY/Au, 4-MPY/Au+Hg^2+^ (50 μg/L) in 0.1 M KCl with 5 mM [Fe(CN)_6_]^3−/4−^. A Randles equivalent circuit used to fit the experimental impedance spectra is shown in the inset.

**Figure 3 micromachines-14-00739-f003:**
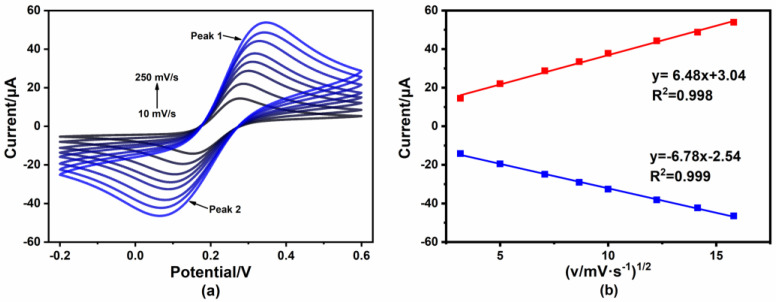
(**a**) CV curves of 4-MPY/Au in 10 μg/L Hg^2+^ (in 0.1 M KCl with 5 mM [Fe(CN)_6_]^3−/4−^) at different scan rates (10–250 mV·s^−1^); (**b**) the relationship between the REDOX peak current and the square root of scan rate.

**Figure 4 micromachines-14-00739-f004:**
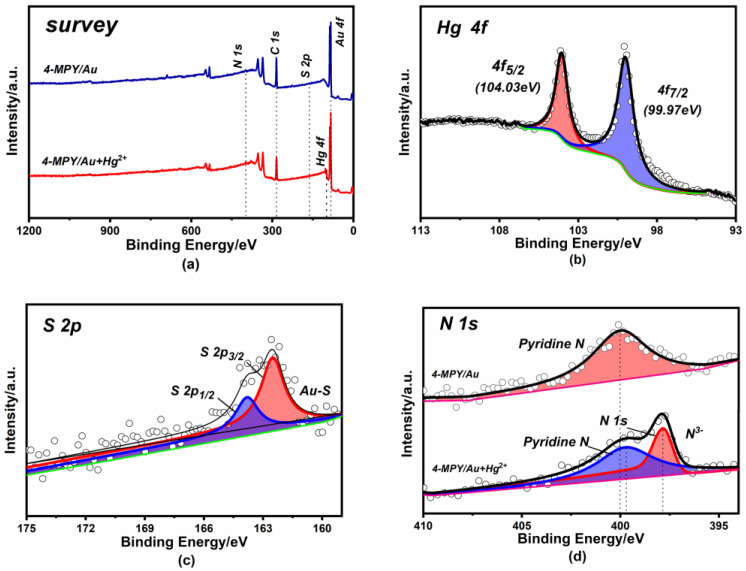
XPS spectra of the modified electrode before and after the reaction with Hg^2+^. (**a**) The survey spectrum; (**b**) Hg 4f; (**c**) S 2p; (**d**) N 1s.

**Figure 5 micromachines-14-00739-f005:**
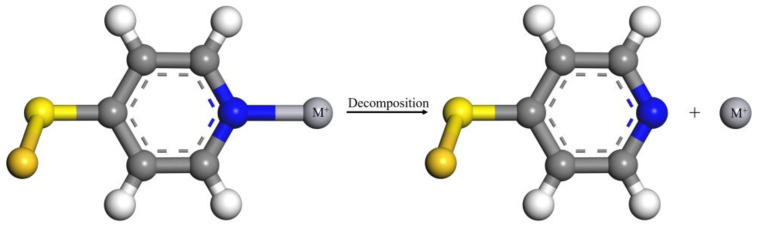
A model for the decomposition reaction of metal ions and pyridine complex(M-Py).

**Figure 6 micromachines-14-00739-f006:**
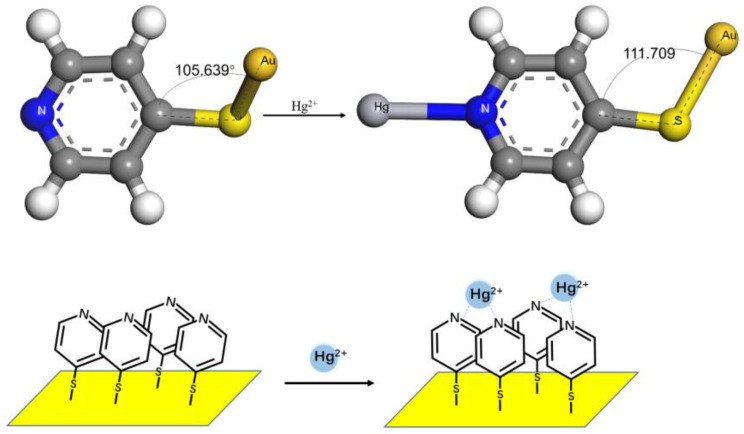
Schematic diagram of Hg^2+^ detection mechanism by using 4-MPY modified Au electrode with the theoretical calculations for the interaction of Hg^2+^ and 4-MPY.

**Figure 7 micromachines-14-00739-f007:**
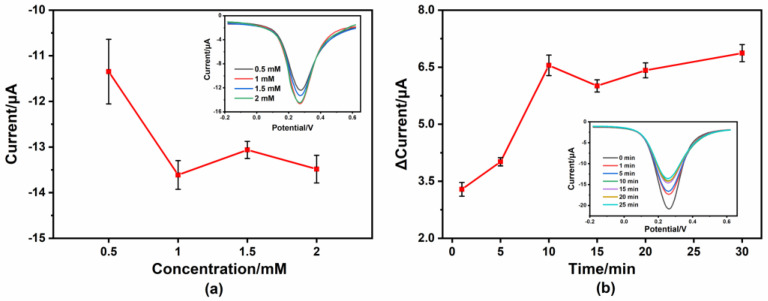
(**a**) The effect of 4-MPY concentration on the current response of the modified electrode by DPV measurement; (**b**) influence of reaction time on the detection of Hg^2+^ (in 5 mM [Fe(CN)_6_]^3−/4−^ with 10 μg/L Hg^2+^).

**Figure 8 micromachines-14-00739-f008:**
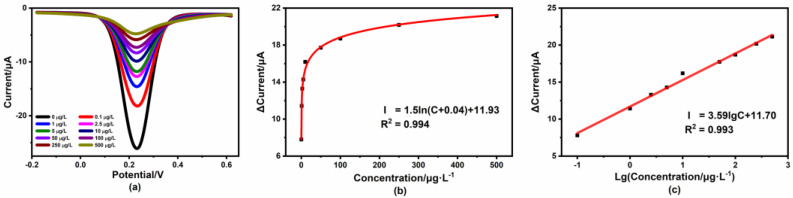
DPV measurements for different Hg^2+^ concentrations. (**a**) DPV plots; (**b**) the relationship between the peak current change and Hg^2+^ concentration; (**c**) calibration curve for peak current change vs. the logarithm of Hg^2+^ concentration.

**Figure 9 micromachines-14-00739-f009:**
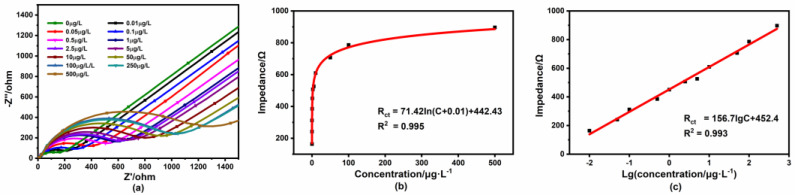
EIS measurements for different Hg^2+^ concentrations. (**a**) Impedance spectra; (**b**) the relationship between the impedance and Hg^2+^ concentration; (**c**) calibration curve for impedance vs. the logarithm of Hg^2+^ concentration.

**Figure 10 micromachines-14-00739-f010:**
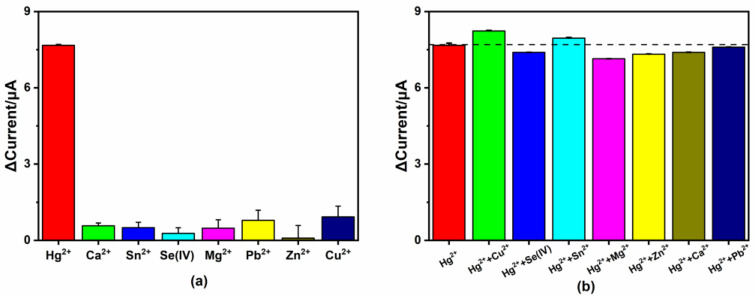
(**a**) The current responses in the selectivity experiment for 4-MPY-modified electrochemical sensor to different interferents (in 5 mM [Fe(CN)_6_]^3−/4−^ with 10 μg/L Hg^2+^ or 100 μg/L other interferent metal ions); (**b**) the current responses in the anti-interference ability experiment of other interferents (in 5 mM [Fe(CN)_6_]^3−/4−^ with 10 μg/L Hg^2+^ and 100 μg/L other interferents).

**Table 1 micromachines-14-00739-t001:** Comparison with different assays via the electrochemical method for Hg^2+^ detection.

Electrode	Sensing Material	Method	Linear Range	LOD	Reference
Glassy carbon electrode	Sr@FeNi-S nanoparticles/carbon nanotubes	DPV	10 μg/L–55.8 mg/L	0.1 μg/L	[26]
Fluorine tin-oxide electrode	Polyamide 6/cellulose/reduced graphene oxide	DPV	500 μg/L–15 mg/L	1 μg/L	[27]
Gold electrode	Guanine nanowire	Chronoamperometry	0.02 μg/L–20 μg/L	0.007 μg/L	[44]
Gold electrode	Carbon Nanotubes/oligonucleotide	SWV ^1^	0.2 pg/L–20 μg/L	0.1 pg/L	[45]
Glassy carbon electrode	Nickel tungstate nanoparticles	DPSV ^2^	2 μg/L–120 μg/L	0.5 μg/L	[46]
Magnetic carbon paste electrode	Fe_3_O_4_/MnO_2_/halloysite nanotubes	DPV	0.5 μg/L–150 μg/L	0.2 μg/L	[47]
Screen-printed carbon electrode	Polypyrrole decorated graphene/β-cyclodextrin	DPV	0.2 μg/L–10.3 mg/L	0.09 μg/L	[48]
Gold electrode	4-Mercaptopyridine	DPV	0.1 μg/L–500 μg/L	0.02 μg/L	This work
		EIS	0.01 μg/L–500 μg/L	0.002 μg/L	

^1^ SWV: square wave voltammetry. ^2^ DPSV: differential pulse stripping voltammetry.

**Table 2 micromachines-14-00739-t002:** Analysis of actual water samples.

	Added (μg/L)	Found (μg/L)	RSD (%, n = 3)	Recovery (%)
Tap water	0.01	0.0094	3.82	94.66%
	0.05	0.049	9.63	98.69%
	10	11.169	0.52	111.69%
Pond water	0.01	0.0092	5.39	92.02%
	0.05	0.059	1.41	118.68
	10	11.12	5.46	111.19%

## Data Availability

The data that support the findings of this study are available from the corresponding authors upon reasonable request.

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
