# Peer review of "4-Mercaptopyridine-Modified Sensor for the Sensitive Electrochemical Detection of Mercury Ions"

_micromachines, 2023, doi:10.3390/mi14040739_

Round 1

Reviewer 1 Report

In this study, the authors have introduced 4-mercaptopyridine (4-MPY) as sensing materials for Hg2+ detection. As a bifunctional molecule, 4-MPY can be self-assembled on the surface of Au electrode through the forming of Au-S bond and coordinate with Hg2+ via nitrogen of the pyridine moiety to form Hg(pyridine)2 (Hg(py)2) complex. The geometric structure of the complex is optimized by density functional theory (DFT) calculation and binding energies (BE) of different ions are calculated. In the presence of Hg2+, Hg(py)2 will accumulate at the sensing region, which leads to reduction of the electrochemical activity of the electrode surface. The concentration of Hg2+ can be detected by analyzing the variation of current or impedance. This proposed electrochemical sensor exhibits excellent sensitivity, selectivity, wide detection range and low LOD. In addition, this sensor has been employed to detect trace Hg2+ in environment water by the standard addition method with satisfactory recovery.

The article is well written and I suggest its publication in a current form.

Reviewer 2 Report

The article “4-Mercaptopyridine modified sensor for the sensitive electrochemical detection of mercury ions” is relevant for scientific community but needs some improvement in some parts.

I have comments, questions and suggestions that can enhance the performance of manuscript, enumerated below:

1.     The key-words must be different from the title to get more visibility for your work.

2.     In lines 37-39, authors should add some references about electrochemical detection using examples for chemical, biological and environmental detection.

3.     The improvement of electrochemical performance of several materials can be used. Please reference some more actual works with these materials

4.     4-MPy is not a conductive material. Why authors have beem proposed this material for electroanalytical applications? The Pros and Cons must appear in the introduction section.

5.     Other metals could also have an interaction with 4-MPy. Explain in order og stability and formation constants why Hg2+ could be more selective for this ligand.

6.     In Fig 2, why the electrode modified with 4-MPy showed more sensitive and showed minor Rc-t than Au electrode.

7.     The paragraph of 148-164 lines is confusing: What about y=11.88x^0.4? In line 161 is ADSORPTION-CONTROLLED process and not ABSORPTION. Authors should appeal to the specialized literature and change all this paragraph that is a messy.

8.     References of XPS peaks must be appear in the text.

9.     In Fig 7, how authors made this study? It must appear the analytical signal too.

10.  Conclusions must be improved.

11.  More actual references should appear in the text too.

Reviewer 3 Report

Authors reported a study on the fabrication of an electrochemical sensor based on 4-mercaptopyridine (4-MPY) self-assembled on gold for the detection of mercury ions in aqueous solutions. They used an indirect method based on the detection of ferrocyanide by the electrode that bound the mercury ions. The paper is interesting and it is informative; thus, I would like to recommend the paper for publication in Micromachines after some important issues are addresses. Here some comments for the authors to improve their manuscript:

English and style must be improved;

I suggest to the authors to point out in the introduction section their innovative analytical method, for the detection of metal ions based on in the aftermath detection of ferrocyanide. They should compare their fabrication method with other reported in literature using a direct analytical method. I suggest to the authors to include in the reference list the following paper:  

Antonino Scandurra, Francesco Ruffino, Mario Urso, Maria Grazia Grimaldi and Salvo Mirabella, Disposable and Low-Cost Electrode Based on Graphene Paper-Nafion-Bi Nanostructures for Ultra-Trace Determination of Pb(II) and Cd(II) Special Issue “Development and Evaluation of Nanostructured Electrochemical Sensors”, Nanomaterials, 2020, 10, 1620; https://doi.org/10.3390/nano10081620.

The authors used gold as support, that is expensive, even if the electrode they proposed is reusable. In the suggested paper it is explained how to obtain simple, low cost, and high-sensitive electrode per the detection of heavy metal ions in aqueous solutions.

Replace in the whole text “drop casted” with “drop cast”; captured with acquired;

Line 28. Use limit of tolerance instead of “standard” term

Lines 93, 94: why the authors used the ferrocyanide? Usually, ferrocyanide reacts with gold and modify the surface of the electrode.

Line 95: use packed instead of “dense”;

Line 99: please explicit the acronym DMol3

Line 111: galvanostatic mode

Lines 152-153. The authors say that “the relationship between current magnitude and scan rate satisfies allometric function, and the equation is ?=11.88?0.4 with a good correlation coefficient of 0.999, which indicates that a diffusion-controlled process is occurring at the surface of the modified electrode”. This conclusion disagrees me. Generally, the current is plotted versus the square root of the scan rate of the potential to see if it satisfies the Randles Sevicik equation and if there is a process controlled by the diffusion.

Line 172. Authors should report the high-resolution Hg 4f XPS spectral region; in the wide spectra of Figure 4a Au 4f and Hg4f seems centered at the same binding energy;

Line 201: Table S2?

Figure 7a: Why does the current oscillate from low values to high? What current is?

Figure 8c is not described in the text; please add a comment;

In Figure S3 negative current what means? Please explain;

In the Selectivity and anti-interference ability studies the potassium is added together the ferrocyanide anion as potassium ferrocyanide at 5 mM. However, the concentration of this ion is reported as 100 micrograms for liter in Figure 10. This should be accounted in Figure 10a,b. Please comment on this;

Line 307 table S2?

 In Table S2 Cu++ and Zn++ have higher binding energy than Hg++, but these ions show a little interference in Figure 10a. How the authors explain the discrepancy between binding energy and interference test?

Round 2

Reviewer 2 Report

Authors have been improved the manuscript and then I recommend to accept the manuscript in Micromachines

Reviewer 3 Report

Authors answered exhaustively and made all the suggested changes. Only one observation concerns the citations of the reference 29 on lines 55-58. References are universally cited as "first author" et al. ..... instead of last author ... say ..... So I invite the authors to change the sentence in lines 55-58 as: "Scandurra et al. proposed a disposable and low-cost electrode based on graphene paper-nafion-Bi nanostructures material for simultaneous determination of cadmium and lead [29]."
